# DetGPT: Detect What You Need via Reasoning

**Renjie Pi[1]\*, Jiahui Gao[2]\*, Shizhe Diao[1]\*, Rui Pan[1], Hanze Dong[1], Jipeng Zhang[1],**
**Lewei Yao[1], Jianhua Han[3], Hang Xu[2], Lingpeng Kong[2], Tong Zhang[1]**

[1]The Hong Kong University of Science and Technology
[2]The University of Hong Kong   [3]Shanghai Jiao Tong University

{rpi,sdiaoaa,rpan,hdongaj,jzhanggr,lyaoak}@ust.hk, sumiler@connect.hku.hk,
xbjxh@live.com, hanjianhua2012@gmail.com,
lpk@cs.hku.hk, tongzhang@tongzhang-ml.org

## Abstract

Recently, vision-language models (VLMs) such as GPT4, LLAVA, and MiniGPT4 have witnessed remarkable breakthroughs, which are great at generating image descriptions and visual question answering. However, it is difficult to apply them to an embodied agent for completing real-world tasks, such as grasping, since they can not localize the object of interest. In this paper, we introduce a new task termed **reasoning-based object detection**, which aims at localizing the objects of interest in the visual scene based on any human instructs. Our proposed method, called **DetGPT**, leverages instruction-tuned VLMs to perform reasoning and find the object of interest, followed by an open-vocabulary object detector to localize these objects. DetGPT can automatically locate the object of interest based on the user's expressed desires, even if the object is not explicitly mentioned. This ability makes our system potentially applicable across a wide range of fields, from robotics to autonomous driving. To facilitate research in the proposed reasoning-based object detection, we curate and open-source a benchmark named **RD-Bench** for instruction tuning and evaluation. Overall, our proposed task and DetGPT demonstrate the potential for more sophisticated and intuitive interactions between humans and machines.

## 1   Introduction

In recent years, the natural language processing field has seen remarkable advancements in the development of increasingly large language models (LLMs). LLMs such as GPT-3 (Brown et al., 2020), Bloom (Scao et al., 2022), PaLM (Chowdhery et al., 2022), Megatron-Turing-530B (Smith et al., 2022), Chinchilla (Hoffmann et al., 2022), and others have expanded the horizons of language understanding and generation. These neural networks, with hundreds of billions of parameters,

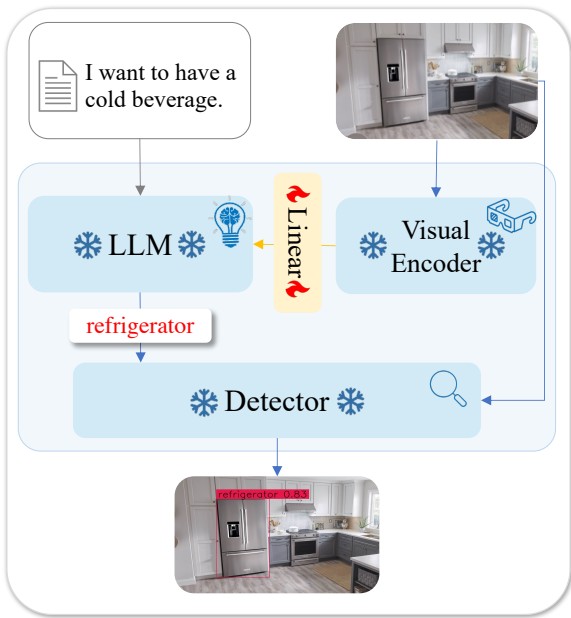

Figure 1: Framework of DetGPT. The VLM consisted of vision encoder and LLM interprets the user instruction, reasons over the visual scene, and finds objects matching the user instruction. Then, the open-vocabulary detector localizes these objects base on the VLM's output.

exhibit human-like proficiency in complex reasoning (Wei et al., 2022; Wang et al., 2022b; Zhou et al., 2022; Zhang et al., 2022; Diao et al., 2023b; Shum et al., 2023). Simultaneously, breakthroughs in the image and multimodal processing, as exemplified by vision-language models (VLMs) like GPT4 (OpenAI, 2023), LLAVA (Liu et al., 2023a) and MiniGPT-4 (Zhu et al., 2023), have empowered the LLMs with ability to understand image inputs. These cutting-edge innovations are highly promising for a diverse array of applications across numerous fields.

As highlighted by recent studies (Shah et al., 2023; Brohan et al., 2022; Fang et al., 2020), since intelligent robot heavily relies on interactions with humans, the field of embodied AI / robotics is set to experience a significant transformation. With the

---
\*Equal Contribution.   Code is available at https://github.com/OptimalScale/DetGPT.

emergence of human-like intelligence of LLMs and VLMs, robots will be able to interpret human instructions and reason over visual scenes, enabling them to execute corresponding actions. This breakthrough will lead to the creation of intelligent robots that are more helpful to humans, and opens up possibilities for various fields.

However, it is important to note that while VLMs have made remarkable progress in generating high-quality image descriptions, this alone is insufficient for robots to interact with the physical world effectively. To achieve this, robots must be able to accurately identify and localize objects within visual scenes, which is a vital prerequisite for performing actions such as "moving" and "grasping" objects. This goal of "localizing objects" is closely linked to the field of object detection, which is one of the most fundamental and extensively studied research areas in computer vision. Conventional object detection systems, such as Faster-RCNN (Ren et al., 2015), Retina-Net (Lin et al., 2017), and YOLO (Redmon et al., 2016) can only detect a fixed number of object categories, which restricts their practicality. Recently, a series of open-vocabulary detection (OVD) systems have emerged as the new trend (Gu et al., 2021; Li et al., 2022; Yao et al., 2022; Liu et al., 2023b). Specifically, those models adopt the contrastive learning approach to align the object-level visual features with the textual class embeddings extracted from a pretrained text encoder (e.g., BERT (Devlin et al., 2019)). In this way, those models are able to detect a much wider range of objects during inference.

Despite the success achieved by OVD systems, they still require humans to provide specific object names, which is neither user-friendly nor realistic. Firstly, human users tend to provide high-level instructions, which may not explicitly contain the object of interest. Secondly, the constraints of human knowledge often hinders the users to provide the object names. For example, the user may wish to identify fruits with a high vitamin K content but lack the necessary expertise to determine which fruits fulfill this requirement. Finally, the range of object categories that humans can supply is intrinsically finite and non-exhaustive. As an illustration, when attempting to detect "objects posing hazards to autonomous vehicles," humans may only be able to enumerate a limited number of scenarios, such as compromised visibility or intricate pedestrian traffic patterns. In summary, it would be desirable

if the detection model is able to interpret human instruction, employ its own knowledge to identify all objects of interest via reasoning, and finally localize them.

To this end, we propose a new research task: **reasoning-based object detection.** In essence, humans provide abstract queries via natural language, then the model discerns and reasons which objects in the image may fulfill the query, subsequently detecting them. We made preliminary explorations in this direction. Specifically, we fine-tune a VLM (e.g., MiniGPT-4 (Zhu et al., 2023)) built on LLMs (e.g., Vicuna (Chiang et al., 2023)) to perform reasoning and predict objects of interest based on user queries (instructions) and input images. We then provide the object names to an open-vocabulary detector for specific location prediction. To facilitate future research in the direction of reasoning-based object detection, we curate a benchmark named **RD-Bench** containing 20000 images and around 120000 query-answer pairs, which will be open-sourced for the research community.

## 2  Related Work

### 2.1  Large Language Models

Recent months have witnessed a transition from encoder-based models (Lu et al., 2019; Devlin et al., 2019; Liu et al., 2019; Jiang et al., 2021; Gao et al., 2022) train seen significant progress in large language models. Models like GPT-3 (Brown et al., 2020), Bloom (Scao et al., 2022), PaLM (Chowdhery et al., 2022), megatron-turing-530b (Smith et al., 2022), and Chinchilla (Hoffmann et al., 2022), have pushed the boundary of language understanding and generation to new frontiers. These massive neural networks have demonstrated human-level abilities in text classification, text generation, knowledge-intensive tasks, and even complex reasoning tasks. Recently, Meta's LLaMA (Touvron et al., 2023) provided a series of powerful open-source models that boost language model research. For example, recent Alpaca (Taori et al., 2023), Vicuna (Chiang et al., 2023), and LMFlow (Diao et al., 2023a) have showcased the powerful capability of instruction-tuned LLaMA.

### 2.2  Vision Language Models

Given the success of language models, many following research explored vision-language interaction, resulting in the development of various multimodal models. The development in a number of

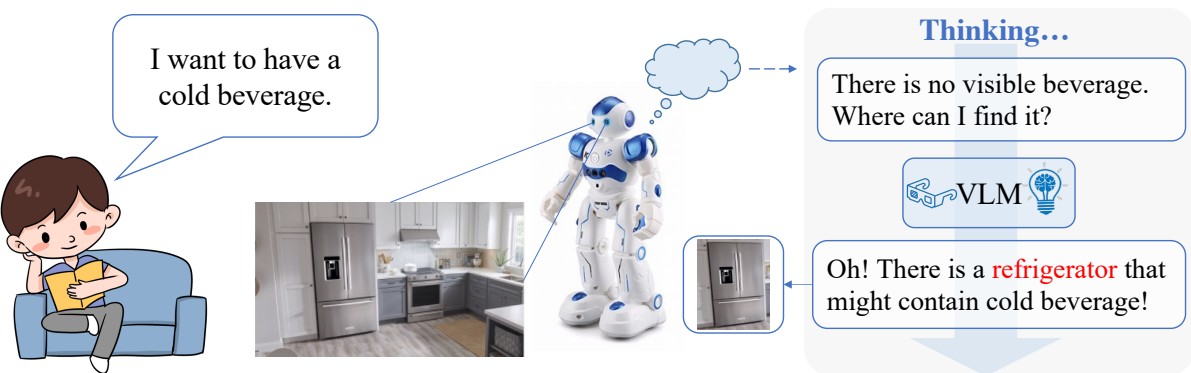

Figure 2: The illustration of reasoning-based object detection task. The detection system is able to interpret human instruction, reason about the visual scene with common sense knowledge, and finally localize the objects of interest.

language model research. Inspired by BERT-like encoder models, most of the multi-modal models (Lu et al., 2019; Tan and Bansal, 2019; Chen et al., 2020; Li et al., 2021; Ding et al., 2021; Li et al., 2020; Ding et al., 2023, 2022) before 2021 are encoder-only Transformers, which are good at cross-modal understanding tasks. However, the transition from encoder-only models to decoder-based models in language model research inspires the pattern shift in multi-modal learning, including encoder-decoder models like VL-T5 (Cho et al., 2021), OFA (Wang et al., 2022a), DaVinci (Diao et al., 2023c), and decoder-only models like GPT-4 (OpenAI, 2023). Most recently, we have witnessed the potential of multi-modal learning due to the powerful language abilities of LLaMA. Recent works include LLaVA (Liu et al., 2023a) and MiniGPT4 (Zhu et al., 2023). Unlike these works, our DetGPT focuses on localizing object of interests based on user instruction, allowing for greater control over objects through language.

### 2.3 Object Detection

Object detection is one of the most fundamental tasks in computer vision, which aims at localizing objects in images. Traditional object detectors have a fixed number of classification heads, which makes them only capable of predicting the classes on which they are trained (Girshick, 2015; Ren et al., 2015; Lin et al., 2017; Yao et al., 2021a; Duan et al., 2019; Yao et al., 2021b; Zhu et al., 2020; Carion et al., 2020). Recently, open-vocabulary object detection has attracted a lot of attention (Gu et al., 2021; Li et al., 2022; Liu et al., 2023b; Yao et al., 2022). The main philosophy is to utilize contrastive training between object visual features and their class embeddings. In such a way, object de-

tectors are able to recognize objects that are unseen during training based on their semantics. Despite the success of open vocabulary object detectors, their ability is still limited in the sense that they can only perform prediction given specific object phrases. On the other hand, our DetGPT enables reasoning and localizing objects of interest given any high-level user instructions.

## 3 Problem Statement

Recent vision language models (VLMs) backed with LLMs have shown promising results in visual understanding based on the visual scene and natural language input. However, they lack the ability for fine-grained visual understanding and precise localization, which makes them difficult to be applied to real-world scenarios involving embodied AI, such as robotics ad autonomous driving. On the other hand, object detection is a crucial task in computer vision, which enables models to analyze images in a fine-grained manner and predict precise object locations. Unfortunately, existing detection methods either can only predict fixed number of object classes, or need exact object names for detection.

To address these limitations, we propose a new task termed **reasoning-based object detection**. In this task, users provide abstract queries using natural language, and the model analyzes both the image and user input, reasons about which objects in the image may fulfill the user's goal, and finally detects their location in the image. For example, as shown in Figure 2, when a user requests "I want to have a cold beverage," the model first analyzes the image of a kitchen and determines that there is no "cold beverage" available. Then, it identifies a refrigerator in the image and, based on the common sense knowledge stored in the LLM, infers that the

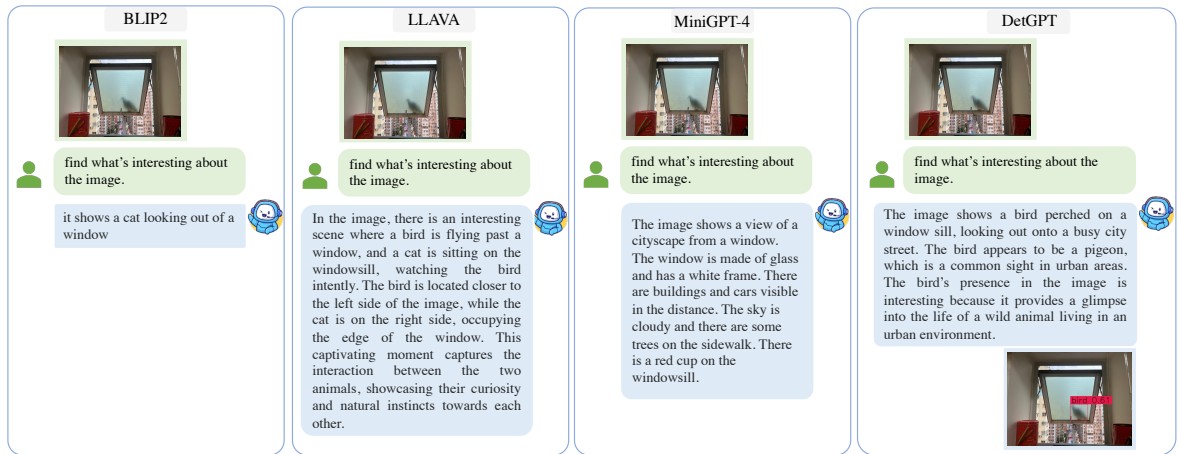

Figure 3: Comparison with other SOTA VLMs. Our DetGPT is able to find and localize the object of interest.

refrigerator may store a cold beverage.

The proposed reasoning-based object detection task and DetGPT open up a world of possibilities for human-machine interactions, which has the potential to greatly improve the capabilities of general-purpose robots.

## 4 Multi-Modal Query-Answer Instruction Data Generation

The traditional way for labelling a dataset requires a lot of human labor. Recently, LLMs such as ChatGPT possess superior generation capability, which can be used to replace human labeling with automatically generated annotations (Schick and Schütze, 2021; Ye et al., 2022a,b; Meng et al., 2022; Gao et al., 2023; Ye et al., 2023). However, such text-only LLMs are not able to interpret visual inputs, which hinders their practicality for data generation based on images.

Motivated by LLAVA (Liu et al., 2023a), we leverage the images of pre-existing datasets (Lin et al., 2014; Shao et al., 2019a), and employ two types of textual annotations to bridge the gap between visual and textual representations: (1) Image Captions, which depict the visual content from different viewpoints. (2) Objects categories, which are the objects present in the image. Specifically, we adopt COCO (Lin et al., 2014) and Objects365 (Shao et al., 2019a) datasets for constructing RD-Bench. Based on the given captions and objects, we design query-answer prompts to instruct ChatGPT to generate the following: (1) a more detailed description of the scene, which gives ChatGPT itself a better sense of the visual scene; (2) query-answer pairs, which consist of a user query (instruction) and the corresponding answer that contains both reasoning process and object names in the image that matches the query. For each image, we generate one detailed description followed by several instruction-answer pairs.

We reorganize the annotations such that each image is associated with the corresponding query-answer pairs. The detailed system prompt for our cross-modal object detection task is shown in Table 9 from the Appendix. To enable better annotation generation, we further manually design two in-context examples for querying ChatGPT, which are shown in Table 10 and Table 11 in the Appendix. Examples of generated detailed description and query-answer pairs are shown in Table 12.

## 5 Method

### 5.1 Model Design

As an initial attempt towards reasoning-based object detection, we propose a two-stage approach that first leverages the VLM to interpret the image and generate relevant objects names/phrases that match the user's instructions via reasoning; then we leverage an open-vocabulary object detector to localize the relevant objects given the results from the VLM. Specifically, for the VLM, we employ a pre-trained visual encoder to extract image features, followed by a cross-modal alignment function to map the image features to the text domain. Then, we utilize a pre-trained LLM as the knowledge brain to interpret both the image features and human instructions, perform reasoning, and determine target objects that help fulfill the given user query. Our framework is illustrated in Figure 1.

Inspired by (Zhu et al., 2023), we employ the

visual encoder of BLIP-2 (Li et al., 2023) as the vision encoder and utilize Vicuna (Chiang et al., 2023) or Robin (Diao et al., 2023a) as the language model to interpret both visual and text features. For the open-vocabulary detector, we leverage Grounding-DINO (Liu et al., 2023b) to localize the target objects in the image. Following MiniGPT-4 (Zhu et al., 2023), we train a linear projection layer from scratch for the cross-modal alignment, which has been proven effective in bridging the gap between vision and language modalities.

**Challenge**   One straightforward approach to implement our proposed framework is combining off-the-shelf VLMs with open vocabulary object detectors without further training. However, we observe that even though with carefully-chosen prompting, VLMs are able to output objects in a specific pattern, they tend to output redundant objects that are either not shown in the image, or unrelated to the user's instruction (shown in Figure 4).

## 5.2  Training and Inference

**Step 1.  Image-Text Pretraining.**   We follow (Zhu et al., 2023) and leverage a combined dataset of SBU, LAION and Conceptual Caption to conduct image-text pretraining. We minimize the language modeling loss:

$$\mathcal{L} = -\sum_{t=1}^{L} \log p \left[ y_t^{i,j} | \mathcal{F}(y_{(<t)}^{i,j}, I^i) \right] \qquad (1)$$

where $\mathcal{F}$ represents the multi-modal model. $I^i$ represents the $i^{th}$ image, and $y_t^{i,j}$ denotes the $t^{th}$ token of the $j^{th}$ answer that belongs to the $i^{th}$ image. $L$ is the length of the answer. During the process, only the projection linear layer is trainable, while all other components are kept frozen. Since the visual feature from BLIP-2 (Li et al., 2023) is already well aligned with textual features, tuning just the linear layer already suffices for aligning with a new language model such as Vicuna.

**Step 2. Instruction Tuning.**   After the first step, although the model is able to generate detailed descriptions based on the image, it is not able to well interpret human instruction and derive objects that fulfill the user's goal. Therefore, we leverage a subset of our curated RD-Bench to fine-tune our model. Similar to the first step, the language modeling loss is optimized, and the linear projection layer is made tunable. We demonstrate that the instruction tuning phase is a crucial step that empowers the VLM to

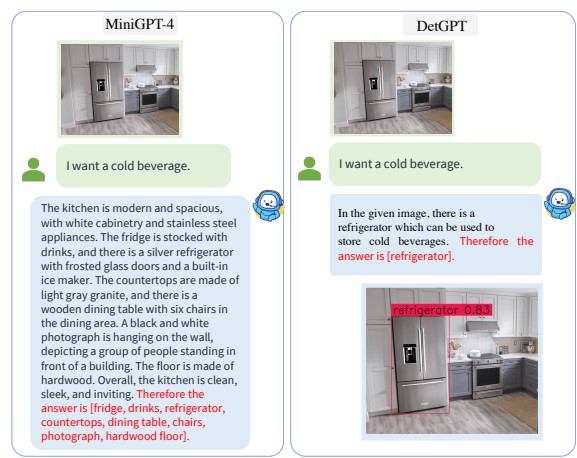

Figure 4: MiniGPT4 vs DetGPT with appended user prompt. The reasoning process and the final sentences containing target objects are both shown for clarity. MiniGPT4 generates redundant objects, while DetGPT can accurately recognize the object of interest.

comprehend fine-grained image features and derive the object of interest given the user's instruction.

Instruction tuning empowers the model to identify the objects of interest in the image. However, we find that the output format of the model often varies, which poses difficulty in extracting the relevant object names/phrases. Therefore, we design a user prompt that is helps the model output the objects strictly in a given format (shown in Table 1), which makes our model more stable. The final input sequence used to train the model is "###Human: ⟨ Img ⟩ ⟨ ImageHere ⟩ ⟨ Img⟩ ⟨ TextHere ⟩ ⟨ User_Prompt ⟩". Blue color represents the input image and Red color represents user instruction.

| **User Prompt** |
| --- |
| Answer me with several sentences.  End the answer by listing out target objects to my question strictly as follows: ⟨Therefore the answer is: [object_names]⟩. |

Table 1: User Prompt. We found that prompting is necessary for listing names of objects of interest in a consistent format, which makes DetGPT more stable.

**Inference.**   During inference, we first provide the model with a system prompt (show in Appendix), which we find to be helpful in stablizing the model's output. Then, we append the user prompt after the user's query. After obtaining the generated answer from the VLM, we extract the object names/phrases from it by matching the specific output format, .i.e, the object names following "Therefore the answer is: ". Finally, we send the

names/phrases and the image to the object detector for localization.

## 6 Demonstration

We present the visualization results of DetGPT in Figure 5 and evaluate its capabilities. Interestingly, DetGPT exhibits the following appealing features: 1) it is proficient in common-sense reasoning based on the user's abstract query and the image; 2) it can utilize the rich knowledge stored in LLMs that are beyond human common sense; 3) thanks to the abundant knowledge stored in the VLM, DetGPT generalizes to a broad range of objects that do not appear during the instruction tuning.

## 7 Experiments

We conduct first stage training on paired image-text data to achieve vision-language alignment. Afterwards, we conduct instruction tuning and evaluation on our curated RD-Bench. Specifically, we randomly sample a subset of 5000 images that are originally from COCO dataset and the corresponding query-answer pairs, which accounts for around 30000 query-answer pairs for instruction tuning. For evaluation, we sample (1) 1000 images that are originally from COCO dataset to evaluate DetGPT's in domain (ID) performance, and (2) 1000 images from Object365 dataset, which are not seen by the model during training, to test its out of domain (OOD) performance.

**Training Details.** For first stage training, the learning rate is set to $1 \times 10^{-4}$, the batch size is set to 128, and the model is trained for 40000 steps. For instruction tuning, the learning rate is $3.5 \times 10^{-5}$, the batch size is set to 32, and the model is trained for 40000 steps. We use adamW as the optimizer, with cosine learning rate scheduler. We use 8 A40 GPUs to conduct all experiments.

**Evaluation.** We conduct evaluation using the conventional metric for object detection, i.e., mean average precision (mAP), which quantifies how well the predicted bounding boxes overlap with the ground truth ones. Specifically, the precision and recall are defined as follows:

$$\text{P} = \frac{TP}{TP + FP} \qquad \text{R} = \frac{TP}{TP + FN}$$

where TP represents true positives (correctly detected objects), FP represents false positives (incorrectly detected objects), FN represents the number of false negatives (missed detections). Then, Average Precision (AP) and Mean Average Precision (mAP) are defined as:

$$\text{AP} = \sum_{r \in R} \text{P}(r) \cdot \Delta\text{R}(r) \qquad \text{mAP} = \frac{1}{N}\sum_{i=1}^{N} \text{AP}_i$$

where R represents the set of recall values, AP can be interpreted as the area under the precision-recall curve. N is the total number of classes, and $\text{AP}_i$ represents the average precision for class i.

Rather than deriving the mAP for all the objects in the image, we calculate only the objects that fulfill the user's queries. Therefore, detecting objects that are irrelevant to the user query will be counted as FP and decrease the final evaluation metric.

There exist two major issues during evaluation: (1) the language model's output can not be guaranteed to exactly match the object names in the benchmark, even though they share the same meaning; (2) there may exist hierarchy among the categories, e.g., a stuffed animal is also a toy. To address the above problems, we first leverage FastText (Joulin et al., 2016) to calculate the similarities between the LLM-predicted objects and all the class names in the benchmark (COCO and Object365 have 80 and 365 classes, respectively). Then, we take the top 1, top 5 and top 10 class names for each LLM-predicted object, and check if the ground truth class of the object is included. This approach makes our evaluation more robust for reasoning-based object detection, since the words that share similar meanings and those that possess hierarchies tend to have higher similarities in their word embeddings.

**Main Results.** As the first attempt in our proposed reasoning-based object detection task, we conduct exhaustive experiments on different variants of DetGPT using our curated RD-Bench. Specifically, we study the effect of instruction tuning, system prompt and user prompt. As shown in Table 2, instruction tuning contributes the most performance gain, and prompts are able to further enhance the model's performance.

**Varying the Language Model.** We conduct experiments on different language model families with various model sizes. Specifically, we adopt Flan-T5 (Chung et al., 2022), Vicuna (Chiang et al., 2023) and Robin (Diao et al., 2023a). As demonstrated in Table 3, Flan-T5 model family achieves inferior performance compared with Vicuna and Robin model families. This result verifies that the

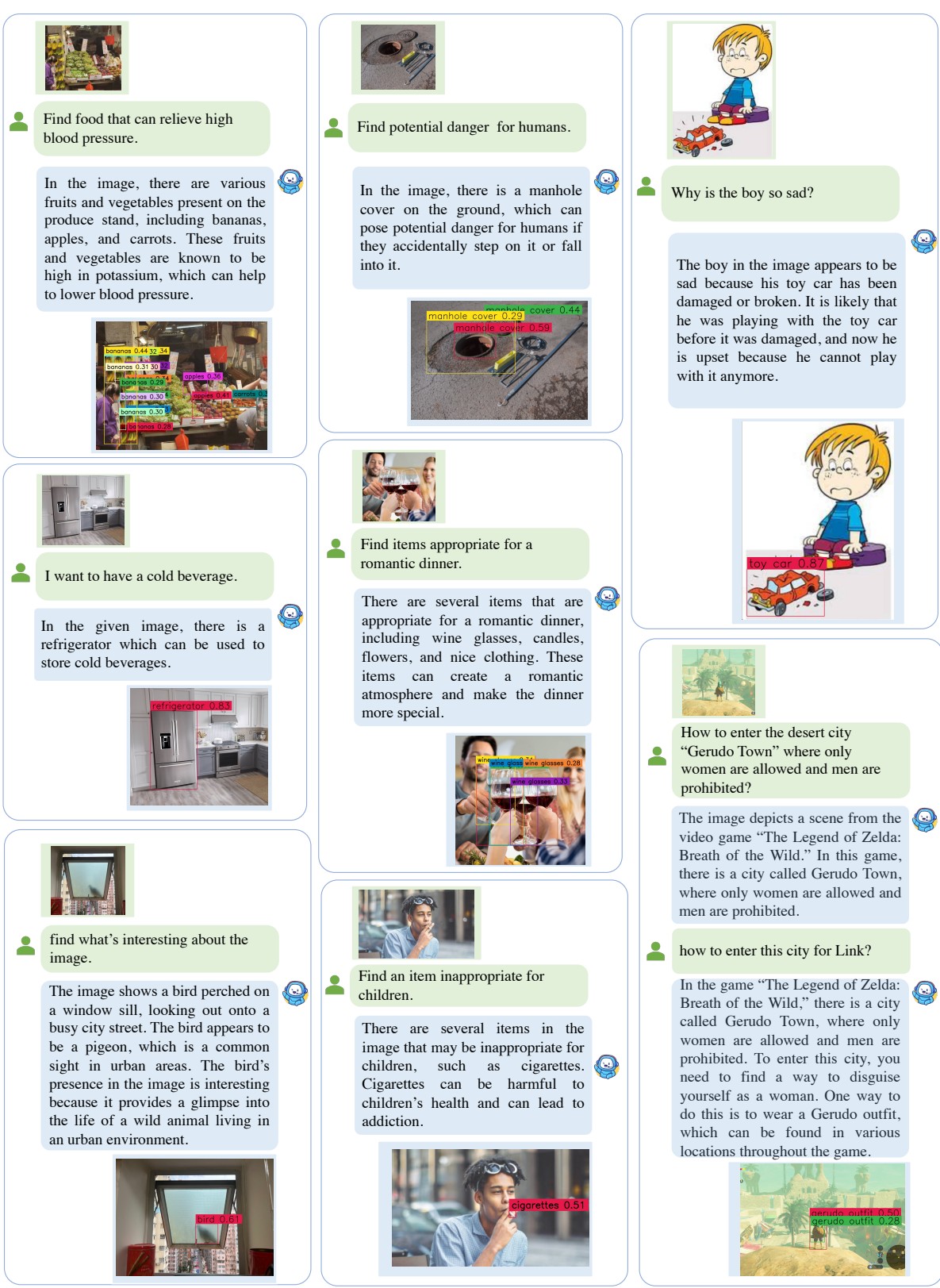

Figure 5: Demonstration of the reasoning process and generated bounding boxes of our DetGPT. Due to space limitation, we do not show the system prompt, the user prompt, or the final sentence "<Therefore the answer is: [object_names]>" for outputting the object names.

| TUNE | SYS | USER | COCO (In Domain) | | | | Objecr365 (Out of Domain) | | | |
|---|---|---|---|---|---|---|---|---|---|---|
| | | | MAP(1) | MAP(5) | MAP(10) | AVG | MAP(1) | MAP(5) | MAP(10) | AVG |
| ✓ | | | 55.26 | 59.22 | 64.50 | 59.66 | 22.51 | 27.95 | 30.03 | 26.83 |
| | ✓ | | 26.24 | 30.30 | 33.08 | 29.87 | 9.65 | 10.91 | 11.38 | 10.65 |
| | | ✓ | 41.16 | 42.20 | 49.88 | 44.41 | 17.95 | 20.35 | 22.21 | 20.17 |
| ✓ | ✓ | | 54.13 | 55.36 | 61.39 | 56.96 | 22.78 | 24.97 | 27.64 | 25.13 |
| ✓ | | ✓ | 60.01 | 61.69 | 65.20 | 62.30 | 23.62 | 28.27 | 31.50 | 27.80 |
| | ✓ | ✓ | 33.70 | 40.32 | 50.80 | 41.60 | 15.16 | 18.05 | 22.16 | 18.46 |
| ✓ | ✓ | ✓ | **60.59** | **62.04** | **65.98** | **62.89** | **23.94** | **28.55** | **32.01** | **28.17** |

Table 2: Test Results on RD-Bench. Instruction tuning and prompting enable DetGPT to achieve promising performances on both in domain and out of domain tasks.

| MODEL | SIZE | MAP(1) | MAP(5) | MAP(10) | AVG |
|---|---|---|---|---|---|
| Flan-T5 | 2.7B | 10.27 | 12.30 | 12.59 | 11.72 |
| | 6.7B | 12.53 | 13.94 | 15.02 | 13.83 |
| Vicuna | 7B | 59.45 | 60.19 | 63.68 | 61.11 |
| | 13B | 60.01 | 61.69 | 65.20 | 62.30 |
| Robin | 7B | 56.32 | 60.21 | 62.73 | 59.75 |
| | 13B | 61.03 | 61.29 | 64.36 | 62.23 |

Table 3: Performance with different language models.

| MODEL | PROMPT | MAP(1) | MAP(5) | MAP(10) | AVG |
|---|---|---|---|---|---|
| Vicuna | ✗ | 52.37 | 55.69 | 58.43 | 55.50 |
| | ✓ | 60.59 | 62.04 | 65.98 | 62.89 |
| Robin | ✗ | 53.19 | 55.31 | 59.06 | 55.83 |
| | ✓ | 61.03 | 61.29 | 64.36 | 62.23 |

Table 4: Adding prompts during instruction tuning.

| DATA | REASON | MAP(1) | MAP(5) | MAP(10) | AVG |
|---|---|---|---|---|---|
| COCO | ✗ | 49.04 | 55.04 | 57.28 | 53.78 |
| (ID) | ✓ | 60.01 | 61.69 | 65.20 | 62.30 |
| Object365 | ✗ | 17.17 | 20.79 | 26.75 | 21.57 |
| (OOD) | ✓ | 23.63 | 28.27 | 31.50 | 27.80 |

Table 5: Effect of adding the reasoning process.

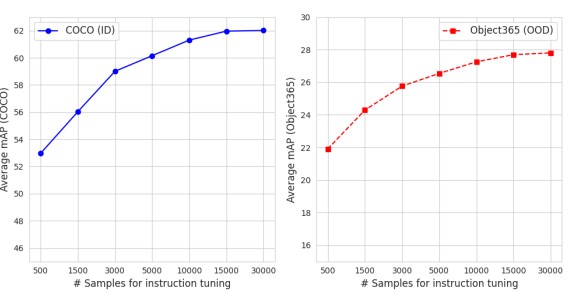

Figure 6: Average mAP for various sizes of instruction tuning datasets. Only a small number of samples are needed for reaching a promising performance.

quality of the language model is crucial for the promising performance of DetGPT.

**Instruction Tuning with Prompting.** As shown in Table 4, we observe that instruction tuning achieves better results if prompts are augmented to the queries during training. This implies that it is desirable to keep the input format consistent during instruction tuning and inference.

**The Impact of Reasoning.** Before outputting the objects of interest, our DetGPT first performs reasoning by describing the image content, then using commonsense knowledge to decide which objects help fulfill the user's query. In Table 5, we analyze the impact of such a reasoning process on the accuracy of detection. Specifically, if we train the model to directly output objects of interest without reasoning, a significant performance drop can be observed. This verifies that reasoning is not only a desirable feature of DetGPT, but also a key factor that helps it accurately derive the object of interest based on human instruction.

**The Impact of Instruction Tuning Size.** From Figure 6, we observe that promising performance can already be achieved with around 10000 sam-

ples for instruction tuning. This verifies that the knowledge stored in the base model is the key to DetGPT's strong performance, and only a small number of samples is needed to empower the model to follow human instructions and output the objects of interest in a standard format.

# 8 Conclusion

We propose a new task termed **reasoning-based object detection**, in which the model needs to interpret high-level human instructions, reason over the visual scene, and finally localize the objects of interest. To facilitate future research in this task, we curate RD-Bench, a dataset that can be used for training and evaluation. Then, we design a two-stage detection pipeline, named DetGPT, which demonstrates strong ability in open-ended tasks and achieves promising performance on our proposed benchmark. We hope our work will pave the way for a more interactive and user-friendly object detection system, which will inspire later works on embodied AI, autonomous driving, and robotics.

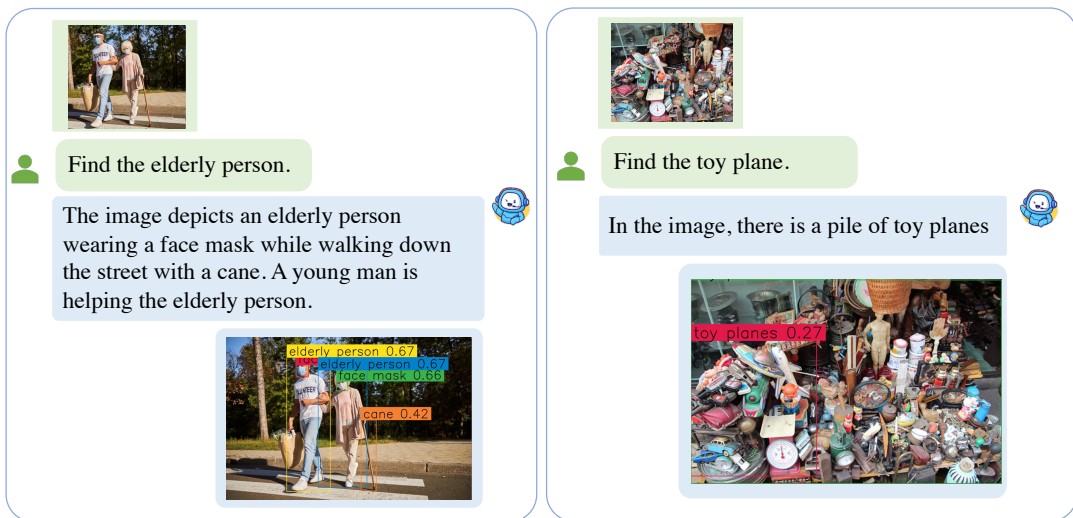

Figure 7: Demonstration of failure cases. Top: even though the multi-modal is able to understand the visual scene and find the elderly lady, the object detector localizes both the young man and the elderly person, and label them both as elderly person. This may be due to the detector is not able to distinguish "young man" from "elderly person". Bottom: there is only one toy plane in the image, but the multi-modal model recognizes "a pile of toy planes". This may be caused by the multi-modal model's lack of fine-grained visual recognition capability.

## 9 Limitation

As the first attempt towards a reasoning-based object detection system, despite the promising results, DetGPT still has some limitations (shown in Figure 7). Due to the two-stage nature of DetGPT, the weaknesses of both open-vocabulary detector and multi-modal models become the bottleneck. For example, we observe that in some cases, even though the multi-modal model is able to find the relevant objects from the image, the open-vocabulary detector is not able to localize them, which may be because the training data of the object detector does not encompass such visual concepts. In some other cases, the multi-modal model is not able to find all relevant objects in the image, possibly due to the lack of fine-grained visual recognition ability. The above limitations promote new research in this direction and demand more advanced solutions.

## 10 Ethic Statement and Broader Impact

The proposed task of reasoning-based object detection and the proposed method, DetGPT, have the potential for significant broader impact across a variety of fields. By enabling an embodied agent to automatically locate objects of interest based on human instructions, DetGPT has the potential to improve the efficiency and effectiveness of tasks such as grasping in robotics and object recognition in autonomous driving. This could lead to safer and more reliable autonomous systems in these fields.

Furthermore, the introduction of RD-Bench as a curated and open-source benchmark for instruction tuning and evaluation can facilitate further research and development in this area, potentially leading to more advanced and versatile applications of reasoning-based object detection.

Overall, the proposed task and method demonstrate a step towards more sophisticated and intuitive interactions between humans and machines.

We do not foresee any ethical concern regarding our paper.

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

# A Data Generation Using ChatGPT

We built our RD-Bench by utilizing images from established datasets like COCO and Object365, as well as the powerful LLM ChatGPT. However, as ChatGPT is designed to handle textual inputs exclusively, we employed caption and bounding box annotations to connect the visual and textual modalities. In the case of Object365, where caption annotations were not available, we employed LLaVA to generate them with the prompt "Describe the image in detail". To showcase the data generation process, we provide both system prompts and contextual examples that were presented to ChatGPT.

**System Prompt** Table 9 displays the system prompt we utilized. In this prompt, we set the role for ChatGPT as a visual assistant to generate query-answer pairs based on the given image. To ensure a comprehensive and diverse range of instructions, we delineated four specific types of queries that needed to be included during data generation. These types include: 1) goal-oriented queries, which associate relevant objects with high-level user queries; 2) detection of all visible objects in the image, similar to conventional object detection tasks; 3) detection of specific objects based on their categories; and 4) attribute-related queries, aimed at localizing objects with specific attributes such as color and shape.

**In-Context Examples** To enable better generation quality, we append two manually written examples after the system prompt as in-context examples, which are shown in Table 10 and Table 11. Specifically, we we first list out the captions and the objects, which are the inputs to ChatGPT. Then we provide a detailed description and the query-answer pairs, which should be the outputs from ChatGPT.

**Generated Data from ChatGPT**   In Table 12, we showcase the inputs and outputs generated by ChatGPT using our proposed data generation pipeline. We have also included the corresponding images for clarity. It is evident from the table that the query-answer pairs generated by ChatGPT are closely aligned with the objects present in the images. This proves the practicality and effectiveness of our data generation pipeline.

## B   System Prompts to DetGPT

We observe that adding a system prompt before the inputs to DetGPT helps stablizing the outputs. Specifically, we show the prompt in Table 13.

## C   Additional Experimets

We conduct additional experiments to validate the superior performance of our model on the reasoning-based object detection task.

| MODEL | TASK 1 | TASK 2 | TASK 3) | TASK 4 |
|---|---|---|---|---|
| GroundingDino | 10.4 | 29.79 | 34.89 | 25.24 |
| DetGPT | 66.34 | 68.53 | 70.26 | 67.22 |

Table 6: Performance comparison with GroundingDino. We demonstrate that our DetGPT is superior at reasoning-based object detection task.

| | FLAMINGO | BLIP-2 OPT6.7B | BLIP-2 FLANT5XXL | DETGPT |
|---|---|---|---|---|
| OK-VQA | 50.6 | 36.4 | 45.9 | 58.5 |

Table 7: Performance comparison on OK-VQA dataset.

| | TASK 1 | TASK 2 | TASK 3 | TASK 4 |
|---|---|---|---|---|
| ViperGPT | 10.37 | 13.43 | 23.64 | 16.21 |
| DetGPT | 58.96 | 59.02 | 61.63 | 60.32 |

Table 8: Performance comparison with ViperGPT.

**Comparison with GroundingDino**   As shown in Table 6, we conduct experiments where we directly use the instructions as input prompts to open-vocabulary detectors, and we pick the bounding boxes by matching the object features with text embeddings of the instructions. In our proposed IOD-Bench, there are many high-level instructions that do not explicitly contain the object names, such as "find all visible objects in the image", and "What should I eat to gain muscle?". In such cases, the obtained bounding boxes may not be associated with an object name/category. Therefore, we turn to a more relaxed criterion for evaluation, where we treat all objects as the "foreground category" and calculate the AP in a class-agnostic manner.

**Comparison with ViperGPT**   We compare our DetGPT with the tool learning-based approach ViperGPT. It is worth noting that ViperGPT lacks support for the "localization" functionality and only generates textual answers. To address this limitation, we extracted the nouns from ViperGPT's answers and subsequently utilized GroundingDino for evaluation purposes. The resulting evaluation scores are presented in the Table below (Due to the requirement of an API key from ChatGPT, we limited our evaluation to a subset of 100 samples to optimize cost efficiency)

**Evaluation on OK-VQA**   We conduct an experiment on the mentioned OK-VQA benchmark and compare the performance with other SOTA methods in Table 7. The results show that our curated RD-Bench dataset indeed also helps boost the performance on open-ended VQA tasks.

**System Prompt for Data Generation**

You are an AI visual assistant, and you are seeing a single image. What you see are a few sentences, which describe the same image you are looking at. Also, some objects that appear in the image are given to you. You are responsible for the following: first, you need to describe the image contents with necessary but not redundant details. then, you need to design queries that can be completed with objects that are shown in the image.

For the query-answer generation, you should generate the following types of queries:
1. goal-oriented queries, where the user inputs something he/she wishes to achieve, and you are responsible to find the objects in the image that helps he/she to do so.
2. detect all objects in the image, where the user wishes to find all the salient objects contained in the image, in this case, you need to find all of them.
3. detect some specific images, where the user wishes to find only a subset of images, such as animals, foods, or electronics, .etc.
4. find objects with certain attribute, such as color or shape.
You should generate as many queries as possible, but the related objects must be contained in the five sentences or the given objects. Also, the queries for an image should be diverse, spanning across all types of queries mentioned above.

Note that you should design attribute-related queries (such as color or shape), only when you are certain about it. Do not generate such queries if the captions provided to you does not contain such information.

You must response any queries or answer in the following way:
Query: ⟨QUERY⟩ Answer: ⟨ANSWER⟩ Therefore the answer is: ⟨TARGET_OBJETCTS⟩

The objects in ⟨TARGET_OBJETCTS⟩ must be shown in the image, and can be used to solve the query in ⟨QUERY⟩.

When answer each query, you must (1) describe all the object (you may refer to the complete object list), (2) based on common sense, use correct object(s) to answer the question, (3) list out target objects in the following manner: "Therefore the answer is: ⟨TARGET_OBJETCTS⟩".

Table 9: System Prompt provided to ChatGPT to generate cross-modal instruction data for reasoning object detection task.

**In Context Example 1**

**Captions:**
Three computer screens sitting on a desk with headphones attached.
A desk that has a few computer screens on it.
A desk with a computer, a laptop, headphones and a telephone office desk with a laptop , headphones, keyboard and dual monitors.
A home computer and keyboard on a desk.

**Objects:**
computer, monitor, keyboard, book, mouse, headphone, telephone

**Description:**
The image depicts a desk with multiple computer screens and various accessories placed on it. Specifically, there are three computer screens present on the desk, each of which has headphones attached to them. In addition to the screens, there is also a laptop, a keyboard, and a telephone present on the desk. The headphones appear to be over-ear style and are connected to the screens via cords. The keyboard is positioned in front of the laptop, and the telephone is located to the left of the screens.

**Query and Answer:**
1.Query: I wish to watch a movie. What can I use? Answer: In the image, there are computer, monitor, keyboard, book, mouse, headphone and telephone. You can use computers and laptops to watch movies online. Therefore the answer is: [computer, monitor]
2.Query: It is late, and I wish to read before going to bed. Answer: In the image, there are computer, monitor, keyboard, book, mouse, headphone and telephone. If you wish to read, you could either read from online sources using the computer, or read from the book. Therefore the answer is: [computer, monitor, book]
3.Query: Find all the computer, mouse, keyboard and cake in the image. Answer: In the image, there are computer, mouse and keyboard, but there is no presence of cake. Therefore the answer is: [computer, mouse, keyboard]
4.Query: I need to call someone. What object can I use in the image? Answer: In the image, there are computer, monitor, keyboard, book, mouse, headphone and telephone. You can use the telephone on the desk to make phone calls. Therefore the answer is: [telephone]
5.Query: What can I use to listen to music? Answer: In the image, there are computer, monitor, keyboard, book, mouse, headphone and telephone. You can use the headphone on the desk to listen to music. Therefore the answer is: [headphone]
6.Query: Detect airplanes and computer in the image? Answer: In the image, there there are computer, monitor, keyboard, book, mouse, headphone and telephone. Therefore, I can detect the computer in the image. However, no airplane can be found. Therefore the answer is: [computer]
7.Query: Detect people and telephone in the image? Answer: In the image, there are computer, monitor, keyboard, book, mouse, headphone and telephone. However, no people can be found. Therefore the answer is: [telephone]
8.Query: Is there any object that is rectangular in shape. Answer: In the image,there are computer, monitor, keyboard, book, mouse, headphone and telephone. The monitors and the laptop are rectangular in shape. Therefore the answer is: [monitor, laptop]
9.Query: Detect computer in the image? Answer: In the image, there are computer, monitor, keyboard, book, mouse, headphone and telephone. Therefore the answer is: [computer]
10.Query: Detect cake and computer in the image? Answer: In the image, there are computer, monitor, keyboard, book, mouse, headphone and telephone. However, no cake can be found. Therefore the answer is: [computer]

Table 10: In-context example.

**In Context Example 2**

**Captions:**
A group of people standing outside of a black vehicle with various luggage.
A person holding a tennis racket about to serve a ball on a tennis court.
A group of people playing tennis on a sunny day.
A tennis player about to serve the ball on a clay court.
Two people playing tennis on a green court with a blue sky in the background.

**Objects:**
tennis racket, tennis ball, tennis court, net, tennis shoes, water abottle

**Description:**
The image shows a tennis court, with a net dividing it in half. A person is standing on one side of the court, holding a tennis racket and getting ready to serve a ball. The court surface appears to be clay, and there are green trees visible in the background. The person is wearing tennis shoes and has a water bottle next to them.

**Query and Answer:**
1.Query: I want to practice my forehand. What object can I use? Answer: In the image, there are tennis racket, tennis ball, tennis court, net, tennis shoes and water abottle. You can use the tennis racket and tennis ball to practice your forehand. Therefore the answer is: [tennis racket, tennis ball]
2.Query: What can I use to hydrate myself during the game? Answer: In the image, there are tennis racket, tennis ball, tennis court, net, tennis shoes and water abottle. Using the water bottle present, you can hydrate yourself. Therefore the answer is: [water bottle]
3.Query: Find all the objects that are used in a tennis game. Answer: In the image, there are tennis racket, tennis ball, tennis court, net, and tennis shoes, all of which are used in a tennis game. Therefore the answer is: [tennis racket, tennis ball, tennis court, net, tennis shoes]
4.Query: I want to play tennis with my friend. Answer: In the image, there is a tennis court, tennis racket, and tennis ball present, which you can use to play tennis with your friend. Therefore the answer is: [tennis court, tennis racket, tennis ball]
6.Query: Find all the objects that are round in shape. Answer: In the image, there are tennis racket, tennis ball, tennis court, net, tennis shoes and water abottle. Among them, the tennis ball is round in shape. Therefore the answer is: [tennis ball]
7.Query: Find the ice cream and the tennis racket. Answer: In the image, there are tennis racket, tennis ball, tennis court, net, tennis shoes and water abottle. However, there is no ice cream in the image. Therefore the answer is: [tennis racket]

Table 11: In-context example.

## Generated Examples

**Captions:**
Some people in a grass field flying a kite in the sky.
A boy playing with a kite in the park.
a colorful kite flying by some people in the park.
A woman and a young girl holding a kite on a green field.
A group of people standing on a field flying a colorful kite.

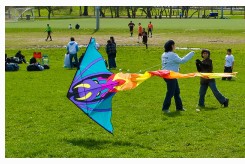

**Objects:**
chair, kite, backpack, person

**Description:**
The image shows a group of people on a grassy field, flying a colorful kite in the sky. Some of the people are standing while others are sitting on chairs. One woman and a young girl are holding onto the kite while it flies in the air. There are trees and other greenery visible in the background.

**Query and Answer:**
1. Query: I want to fly a kite. What object do I need? Answer: In the image, there is a kite present, which you can use to fly a kite. Therefore the answer is: [kite]
2. Query: Find all the people in the image. Answer: In the image, there are multiple people visible, standing and sitting, while flying a kite. Therefore the answer is: [person]
3. Query: I want to sit while flying a kite. What object can I use? Answer: In the image, there are chairs present on the grassy field, which you can use to sit while flying a kite. Therefore the answer is: [chair]
4. Query: Find all the objects that are colorful. Answer: In the image, there are multiple objects visible, including the kite flying in the sky, which is colorful. Therefore the answer is: [kite]
5. Query: Find all the objects that are used for carrying things. Answer: In the image, there is a backpack visible, which is used for carrying things. Therefore the answer is: [backpack]

**Captions:**
Two computers are sitting on top of the desk.
Two computers on a large wooden computer desk.
A desk with a keyboard, laptop and monitor.
A computer monitor next to a keyboard, laptop and a mouse.
A laptop with external keyboard, mouse, phone and photo on a desk.

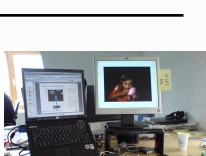

**Objects:**
chair, person, tv, cell phone, cup, laptop, mouse, keyboard

**Description:**
The image displays a desk with various computer accessories on it. There are two computers present on the desk, which appear to have external keyboards attached to them. In addition to the computers, there is a laptop, computer monitor, and mouse present on the desk. A phone is visible as well, along with a photo frame. The desk appears to be made of wood, and there are no other visible objects or people in the scene.

**Query and Answer:**
1. Query: How can I make a phone call? Answer: In the image, there are keyboard, person, chair, laptop, mouse, cup, cell phone, and TV. To make a phone call, use the cell phone on the desk. Therefore the answer is: [cell phone]
2. Query: Find all the electronic devices in the image. Answer: In the image, there are keyboard, laptop, mouse, computer monitor, cell phone, and TV. All of these are electronic devices. Therefore the Answer: is: [keyboard, laptop, mouse, computer monitor, cell phone, TV]
3. Query: Find all the objects that can be used for typing. Answer: In the image, there are keyboard, laptop, and external keyboard. All of these can be used for typing. Therefore the answer is: [keyboard, laptop, external keyboard]
4. Query: Can you find any object that is black in color? Answer: In the image, there are keyboard, person, chair, laptop, mouse, cup, cell phone, and TV. The external keyboard and computer mouse are black in color. Therefore the answer is: [external keyboard, mouse]
5. Query: Detect all the objects that are rectangular in shape. Answer: In the image, there are keyboard, laptop, mouse, computer monitor, cell phone, and TV. Among them, the computer monitor and the laptop can be considered rectangular in shape. Therefore the answer is: [computer monitor, laptop]

Table 12: Two examples to demonstrate the instruction-following data. The top section displays the prompts used to instruct text-only ChatGPT, which consists of captions and objects of the visual image. The bottom section shows the responses generated by ChatGPT.

| **System Prompt for DetGPT inference** |
| --- |
| You must strictly answer the question step by step:
Step-1. describe the given image in detail.
Step-2. find all the objects related to user input, and concisely explain why these objects meet the requirement.
Step-3. list out all related objects existing in the image strictly as follows: ⟨ Therefore the answer is: [object_names] ⟩.
Complete all 3 steps as detailed as possible.
You must finish the answer with a complete sentence. |

Table 13: System Prompt provided to DetGPT during inference. We found that prompting is necessary for listing names of object of interest in a consistent format, which makes DetGPT more stable.