# OpenReview forum: "DetGPT: Detect What You Need via Reasoning"
_EMNLP/2023/Conference — EMNLP 2023 Main_

### Official Review · Reviewer_Hwbg · 2023-07-25

**Typos Grammar Style And Presentation Improvements:** Format issue
**Soundness:** 3

**Excitement:**

3: Ambivalent: It has merits (e.g., it reports state-of-the-art results, the idea is nice), but there are key weaknesses (e.g., it describes incremental work), and it can significantly benefit from another round of revision. However, I won't object to accepting it if my co-reviewers champion it.

**Paper Topic And Main Contributions:**

The paper designs a task named reasoning-based object detection and introduces a method called DetGPT. Vision-language models (VLMs)
 are employed to interpret human instructions and reason about the visual scene, followed by an open-vocabulary object detector to localize the objects of interest.

**Questions For The Authors:**

A. What is the computational cost for different types of architecture in your experiments? It seems larger models gain better results, yet the cost is not clear.
B. Why do you choose ChatGPT instead of text-davinci-003/002 for data generation?

**Reasons To Accept:**

1. Well organized motivation and interesting setting.
2. Comprehensive results with different language models as backbones.

**Reasons To Reject:**

1. A major concern is the annotation quality. An annotator requires multi-modal resources to spot the target object, and the image descriptions may miss important visual information.
2. Format issue: the content exceeds 8 pages (Figure 7)

**Reproducibility:**

3: Could reproduce the results with some difficulty. The settings of parameters are underspecified or subjectively determined; the training/evaluation data are not widely available.

**Reviewer Confidence:**

5: Positive that my evaluation is correct. I read the paper very carefully and I am very familiar with related work.

---

> ### Author Rebuttal · Authors · 2023-08-29
>
> We sincerely thank Reviewer Hwbg for the positive recommendation and we are grateful for the time you spent on our submission. Below we would like to give detailed responses to each of your comments.
>
> **Q1. "A major concern is the annotation quality. An annotator requires multi-modal resources to spot the target object, and the image descriptions may miss important visual information. "**
>
>
> Thank you for pointing out this concern. We wish to clarify that for task tuning data generation,  we aim to generate query-answer pairs for the 2nd-stage instruction tuning, which helps the VLM learn to understand user instructions and detect objects of interest.  Since the query itself is generated to find the object of interest,  fine-grained visual information is not necessary for this purpose. More specifically, we provide the ChatGPT with
> - (1) list of object categories that exist in the image and
> - (2) image descriptions from different perspectives.
>
> Since the query itself is most related to the object of interest, rather than the visual details of the image, the information we provide to ChatGPT is sufficient for generating query-answer pairs.
>
> In addition, we also want to note that we conduct vision-language alignment to equip the VLM with the ability to interpret detailed image features during the 1st-phase pretraining, where we use human-annotated image-caption datasets.  This can be verified as DetGPT achieves satisfactory performance even with objects that do not appear in our 2nd-stage instruction data. For instance, it can successfully identify cracks in walls when prompted to detect potential dangers to humans.  As we are unable to attach figures during rebuttal, we will include additional analysis in our final version.
>
> **Q2. ''Format issue: the content exceeds 8 pages (Figure 7)''**
>
> Thanks for pointing out this issue. We apologize for the confusion.  Figure 7 belongs to the limitation section. We will attach it to the appendix in our final version.
>
> **Q3. ''What is the computational cost for different types of architecture in your experiments?''**
>
> We compare the inference cost of DetGPT with different architectures in terms of frame per second (FPS). We wish to note that the inference cost is mainly introduced by the LLM. Our DetGPT is much faster than ViperGPT[2], which is a tool learning framework based on code generation.
>
>
> |DetGPT-FlanT5-2.7B|DetGPT-FlanT5-6.7B |DetGPT-Vicuna7b | DetGPT-Vicuna13b  | DetGPT-Robin7b  | DetGPT-Robin13b  | ViperGPT  |
> |----------|----------|----------|----------|----------|----------|----------|
> |   1.0 FPS |   0.5 FPS  |   0.4FPS |    0.2FPS  |   0.4FPS |  0.2FPS |  0.05FPS |
>
>
> **Q4. ''Why do you choose ChatGPT instead of text-davinci-003/002 for data generation?''**
>
> In our study, we follow LLaVA [1] to use ChatGPT for data generation, which offers a reliable approach.  Besides, ChatGPT and text-davinci-003 are the advanced RLHF versions of text-davinci-002, providing more informative responses. **An advantage of using ChatGPT is its lower cost in comparison to text-davinci-003, making it a more cost-effective option for our research**.
>
>
> | model | cost |
> |----------|----------|
> | ChatGPT  | $0.0015 / 1K tokens   |
> |  text-davinci-003 | $0.0200 / 1K tokens |
>
> **Q5: Reproducibility**
>
> Thanks for the feedback. We have include the prompts, implementation details in our paper, and upload code as supplementary materials to this submission. We assure we will make the trained model, training/evaluation dataset, and codes publicly available upon acceptation.  As acknowledged by MiAP, our prompt details in the appendix help with reproducibily, and the RD-Bench dataset is a resource contribution to the community. With the released codes and datasets, it is easy to reproduce our results.
>
> [1]LLaVA: Visual Instruction Tuning
>
> [2]ViperGPT: Visual Inference via Python Execution for Reasoning

---

### Official Review · Reviewer_MiAP · 2023-07-26

**Soundness:** 4

**Excitement:**

4: Strong: This paper deepens the understanding of some phenomenon or lowers the barriers to an existing research direction.

**Paper Topic And Main Contributions:**

This paper presents a new dataset (RDBench) and approach (DetGPT) for detecting objects with indirect queries. DetGPT resolves queries like “I want a cold beverage” by prompting an LLM and fine-tuning a VLM to find relevant objects such as a “fridge’’. Evaluations are conducted on RDBench test sets, which consist of both in-domain (COCO) and out-of-domain (Object365) images. The experiments investigate the effectiveness of different LLMs (Flan-T5, Vicuna, Robin), and the importance of LLM-based reasoning.

**Questions For The Authors:**

Summarizing questions from above:
- A: How does a majority-class baseline perform?
- B: Does training on indirect queries reduce performance on direct queries?
- C: Did the authors experiment with chain-of-thought prompting?
- D: Suggestion – Did the authors experiment with multi-step chain-of-thought (Wei et al. 2022) reasoning? Allowing multiple steps for LLM reasoning could help boost performance.

**Reasons To Accept:**

+ Overall, object-detection with indirect queries is an interesting research problem for the vision-language community. Most object detectors assume a predefined list of classes; even open-vocabulary detectors assume that targets are directly specified in the query. Dealing with ambiguous in-direct queries is a challenging problem for vision-language grounding.
+ The experiments include a good set of sensitivity analyses and investigations with various SoTA models. Flan-TF, Vicuna, and Robin are studied extensively. The effectiveness of LLM reasoning, prompting, and dataset sizes are also studied.
+ RDBench makes a resource contribution to the community. Particularly, the recently popular LLM+Robotics community might be interested in the dataset as a standard evaluation platform.
+ Further details on the prompts used are provided in the appendix. This helps with reproducibility.

**Reasons To Reject:**

- No majority-class baseline? In the qualitative examples in Figure 5, the target objects are already apparent and salient in the visual frame. Perhaps running a standard object-detector (MaskRCNN), or an open-vocabulary detector (MDETR) with generic prompts, and picking the largest/most-confident detection in the frame, would be a good baseline?
- Does training on RDBench reduce performance on normal object-detection that does not involve indirect queries? For instance, would DetGPT struggle with a direct query like “the fridge”? This might be worth investigating because DetGPT’s formulation assumes all queries are ambiguous, but real-world interactions could involve a mix of both direct and indirect queries.

References:
We et al. 2022 – https://arxiv.org/abs/2201.11903

**Reproducibility:**

4: Could mostly reproduce the results, but there may be some variation because of sample variance or minor variations in their interpretation of the protocol or method.

**Reviewer Confidence:**

4: Quite sure. I tried to check the important points carefully. It's unlikely, though conceivable, that I missed something that should affect my ratings.

---

> ### Author Rebuttal · Authors · 2023-08-29
>
> We sincerely thank Reviewer MiAP for the positive feedback as well as the valuable suggestions. We are also glad for the acknowledgment that the problem we are working on is an interesting and challenging research problem for the vision-language community. Below we would like to give detailed responses to each of your comments.
>
> **Q1.  ''How does a majority-class baseline perform? ''**
>
> Thanks for the great advice. We believe by “majority-class”, you are referring to the classification datasets where images may contain only one salient object (e.g.,ImageNet). However, we wish to note that the images in COCO and Object365 datasets usually contain many ground truth objects, and the instructions in RD-Bench may be associated with any of those objects. For instance, the fridge example in Figure 5 contains a fridge, an oven, a range hood, a cabinet, and a sink. If the instruction “I want to have a beverage” is given, any predicted objects other than the fridge will be considered false positives and decrease the AP score.
>
> As per your suggestion, we conduct the 2 following experiments on the COCO partition of RD-Bench:
>
> (1) We pick the major category for each image in the COCO dataset and input it to GroundingDino with a generic prompt. Specifically, we utilize CLIP to calculate the similarity score between each image and all 80 category names and pick the category with the largest score as the majority class. The results are as follows:
> |             |  Task 1 | Task 2  | Task 3 | Task 4 |
> |----------|----------|----------|----------|----------|
> |             |   goal-orienated | all obejcts  | specific object|object w certain attribute |
> |      GroundingDino     |   13.29   |  12.57 | 23.03    |  13.48   |
> |      DetGPT      | 59.36   |   63.24   |   66.57   |  59.99    |
>
> We present the task definition and an example for each task below.
> ```
> TASK 1: goal-orientated task,  to find the objects that can finish the user’s goal
> Query: I want to increase my muscle mass.
> Answer: Here are donuts and nuts on the table. Nuts are a good source of protein and contain all essential amino acids needed for muscle growth and repair.
>
>
> TASK 2: detect all objects in the image
> Query: Find all the objects in the image.
> Answer: In the image, there are donuts, nuts, oranges, a bowl and a knife.
>
>
> TASK 3. detect some specific objects
> Query: Find all the knife in the image.
> Answer: In the image,  a knife can be found.
>
>
> TASK 4. find objects with certain attribute, such as color or shape
> Query: What is the object in the image that is yellow?
> Answer:  In the image, there are oranges visible. Oranges are yellow.
> ```
>
>
>
> (2) We conduct experiments where we directly use the instructions as input prompts to open-vocabulary detectors, and we pick the bounding boxes by matching the object features with text embeddings of the instructions. In our proposed IOD-Bench, there are many high-level instructions that do not explicitly contain the object names, such as “find all visible objects in the image”, and “What should I eat to gain muscle?”. In such cases, the obtained bounding boxes may not be associated with an object name/category. Therefore, we turn to a more relaxed criterion for evaluation, where we treat all objects as the “foreground category” and calculate the AP in a class-agnostic manner.
>
>
> |             |  Task 1 | Task 2  | Task 3 | Task 4 |
> |----------|----------|----------|----------|----------|
> |             |   goal-orienated | all obejcts  | specific object|object w certain attribute |
> |      Grounding Dino       |   10.04   |   29.79|  34.89    |  25.24   |
> |      DetGPT      |   66.34   |   68.53   |   70.26   |  67.22   |
>
>
> We believe that the above experiments verify the challenges of our proposed reasoning-based object detection and the effectiveness of DetGPT. We will add those experiments to the updated version of our paper.
>
> **Q2. ''Does training on RDBench reduce performance on normal object-detection that does not involve indirect queries?''**
>
> Thanks for the comment. We wish to clarify that when constructing the dataset, we do not only assume indirect queries, but also direct queries that are already supported by conventional object detection. As shown in “System Prompt for Data Generation”, we incorporate four types of tasks, namely goal-oriented tasks, detecting all visible objects, detect specific objects, and detect object with certain attributes (illustrated below). We believe you are referring to the  “detect specific objects”  task, which is also supported by regular open-vocabulary object detectors. To verify the effectiveness of DetGPT on the mentioned task, we compare it with GroundingDino and on the task and report the AP(5) . For the evaluation of open-vocabulary detectors, the object categories are provided with generic text prompts. We can see that the result is on par with directly using GroundingDino.
>
>
> |             |  DetGPT | Grounding Dino  |
> |----------|-----------|----------|
> |    direct query   (generic text prompts) |67.08   |   67.32|
>
> **Q3. "Chain-of-thought prompting.  Did the authors experiment with multi-step chain-of-thought (Wei et al. 2022) reasoning? Allowing multiple steps for LLM reasoning could help boost performance. "**
>
> We appreciate the suggestion regarding the incorporation of multi-step chain-of-thought prompting. It's worth noting that we have carefully considered and implemented the multi-step chain-of-thought prompting in the DetGPT inference system, as outlined in Table 10. The results in Table 2 validate the suggestion that by incorporating cot prompting, we can achieve even greater enhancements in the performance of detgpt. To provide a clear illustration, we also include the results and the COT prompt in additional tables below:
>
>
> Test results on COCO (In Domain) :
> | COT Sys Prompt | Tune| User | mAP(1) | mAP(5) | mAP(10) | Avg |
> |------|-----|------|--------|--------|---------|-----|
> |          |  ✔️    | ✔️   | 60.01  | 61.69  | 65.20   | 62.30  |
> |  ✔️   | ✔️   | ✔️   | **60.59**  | **62.04**  | **65.98**   | **62.89**  |
>
>
>
>
> Test results on Object 365 (Out of Domain):
> | COT Sys Prompt | Tune | User | mAP(1) | mAP(5) | mAP(10) | Avg |
> |------|-----|------|--------|--------|---------|-----|
> |         |     ✔️| ✔️   | 23.62  | 28.27  | 31.50   | 27.80  |
> |  ✔️   | ✔️   | ✔️   | **23.94**  | **28.55**  | **32.01**   | **28.17**  |
>
>
> Our COT inference system prompt is:
> ```
> You must strictly answer the question step by step:
> Step-1. describe the given image in detail.
> Step-2. find all the objects related to user input, and concisely explain why these objects meet the requirement.
> Step-3. list out all related objects existing in the image strictly as follows: ⟨ Therefore the answer is: [object_names] ⟩.
>  Complete all 3 steps as detailed as possible. You must finish the answer with a complete sentence
> ```

---

### Official Review · Reviewer_5jA8 · 2023-08-05

**Soundness:** 4

**Excitement:**

4: Strong: This paper deepens the understanding of some phenomenon or lowers the barriers to an existing research direction.

**Paper Topic And Main Contributions:**

This paper introduces a new task called reasoning-based object detection, which aims to localize objects in images given language instructions, especially high-level instructions. A framework DetGPT is introduced to solve this task by breaking it down to training a VLM to generate relevant object names based on image inputs and using a pre-trained open-vocabulary object detector to localize the objects based on the predicted object names. A dataset is introduced to train the model and evaluate its performance. To create the dataset, ChatGPT is used to generate additional textual descriptions and question-answering pairs for images in COCO and Object365.

**Questions For The Authors:**

A. Please provide a more detailed analysis of the dataset.

B. Please discuss and provide results for the proposed method on existing benchmarks or compared to existing methods.

**Reasons To Accept:**

1. The task of localizing objects in images based on abstract and high-level language descriptions is interesting.
2. The proposed method can potentially be integrated into a robotic system, where relevant objects are detected based on user instructions or high-level task goals.
3. The authors provide many qualitative examples that demonstrate the unique ability of the proposed method.

**Reasons To Reject:**

1. Since the dataset is automatically generated, we do not know what type of reasoning or commonsense knowledge is required. The authors should provide a more detailed analysis of the dataset and a breakdown of the model performance based on different types of questions or different difficulties of the questions.
2. It’s hard to judge the significance of the results because the evaluation does not involve any known methods or existing benchmark datasets. Since this paper introduces a new task, it’s possible that no baselines are directly applicable. However, the authors could compare individual components of the system to other existing methods. For example, for predicting object names based on images, the authors could compare to well-performing methods on standard benchmarks such as OK-VQA.
3. The authors should provide more evidence on the advantage of the trained VLM over existing LLMs to can generate programs to compose vision-and-language models, such as ViperGPT and VISPROG. I imagine that these methods can first exhaustively detect all objects in an image and leverage LLMs to determine the object that is relevant to the user instruction.
4. There is limited theoretical contribution since the network design and training procedure of the VLM model mainly follows previous work miniGPT4.

**Reproducibility:**

3: Could reproduce the results with some difficulty. The settings of parameters are underspecified or subjectively determined; the training/evaluation data are not widely available.

**Reviewer Confidence:**

3: Pretty sure, but there's a chance I missed something. Although I have a good feel for this area in general, I did not carefully check the paper's details, e.g., the math, experimental design, or novelty.

---

> ### Author Rebuttal · Authors · 2023-08-29
>
> We sincerely thank Reviewer 5jA8 for the review and are grateful for the time you spent with our submission. We are glad for the acknowledgment that our proposed task is interesting and can potentially be integrated into a robotic system. We wish to address your concerns by giving detailed responses to each of your comments as follows:
>
> **Q1. "Detailed analysis of the dataset and a breakdown of the model performance"**
>
> Thank you for the great question, we agree that more analysis of dataset and sub-tasks is needed, and we provide them below. As defined in the system prompt, we have developed four specific task types. Below, we provide the definitions and examples for each task type as follows:
>
> ```
> TASK 1: goal-oriented task,  to find the objects that can finish the user’s goal
> Query: I want to increase my muscle mass.
> Answer: Here are nuts on the table. Nuts are a good source of protein and contain all essential amino acids needed for muscle growth and repair.
>
> TASK 2: detect all objects in the image
> Query: Find all the objects in the image.
> Answer: In the image, there are donuts, nuts, oranges, a bowl and a knife.
>
> TASK 3. detect some specific objects
> Query: Find all the knife in the image.
> Answer: In the image,  a knife can be found.
>
> TASK 4. find objects with certain attribute, such as color or shape
> Query:What is the object in the image that is yellow?
> Answer:  In the image, there are oranges visible, which are yellow in color.
> ```
>
> Notably, task 2, 3 and 4 are included to support direct queries, which are also supported by common object detectors. On the other hand, queries in task 1 contain high level user instructions, which typically aim at achieving a certain goal and do not explicitly mention the object. Among these tasks, task 1 requires common sense reasoning closely related to daily life, which is also the most challenging task.
>
>
> We further break down the percentage of instructions for different tasks:
> |  Task 1 | Task 2  | Task 3 | Task 4 |
> |----------|----------|----------|----------|
> |   goal-orienated | all obejcts  | specific object|object w certain attribute |
> |   37%   |   15%   |   26%   |  22%   |
>
> The instructions of the goal-oriented task has the majority number of query-answer pairs. In the following table, we further evaluate our model’s performances on each one of the four tasks separately. mAP(5) metric is reported. As can be observed, DetGPT is able to perform well across all tasks after task tuning and with appropriate prompts. Meanwhile, task 1 (goal-oriented task) shows slightly lower performance, which verifies that it is more challenging due to the need of reasoning.
>
>
> |  Tune  |   Sys  |  User  |  Task 1 | Task 2  | Task 3 | Task 4 | avg|
> |----------|----------|----------|----------|----------|----------|----------|----------|
> |   ✔️  |    |    |55.36   |   60.82   |   62.31   |  58.39   |  59.22    |
> |     |  ✔️  |    |16.70   |   29.63   |   35.54  |  39.36   |  30.30    |
> |     |     | ✔️   |34.63   |   41.17   |   43.25   |  49.75   |  42.20    |
> |   ✔️  |  ✔️  |    |54.30   |   58.69   |   59.21   |  49.24   |  55.36    |
> |   ✔️  |    |  ✔️  |59.13   |   62.96   |   66.83   |  57.84   |   61.69    |
> |      |  ✔️   | ✔️   |33.75   |  42.03   |   43.52  |  41.98  |   40.32   |
> |  ✔️   |  ✔️  | ✔️   |59.36   |   63.24   |   66.57   |  59.99   |   62.04    |
>
>
> **''What type of reasoning or commonsense knowledge is required.''** Our RD-Bench is based on common objects found in COCO images, including food, ​​vehicles, animals, kitchen tools and more. The current benchmark assesses the reasoning ability of utilizing common sense knowledge in daily life to help users achieve their goals, such as understanding tool usage and the nutritional components of various fruits. Surprisingly, we have discovered that DetGPT achieves satisfactory performance even with objects not explicitly mentioned in our instruction data. For instance, it can successfully identify cracks in walls when prompted to detect potential dangers to humans. This generalization ability is made possible by the rich knowledge contained in the LLM.
>
> **Q2: "Evaluation on OK-VQA"**
>
> We agree that despite the absence of a baseline for the new task, the vision language model component of DetGPT can be evaluated on the VQA datasets. As suggested, we conduct an experiment on the mentioned OK-VQA benchmark and compare the performance with other SOTA methods in the following table. The results show that our curated RD-Bench dataset indeed also helps boost the performance on open-ended VQA tasks.
>
> |             |  Flamingo80B |  BLIP-2 OPT6.7B  | BLIP-2 FlanT5XXL | DetGPT |
> |----------|----------|----------|----------|----------|
> |OK-VQA |  50.6   |   36.4  |   45.9   |  58.5   |
>
> We wish to note that our proposed reasoning based object detection task not only requires identifying objects of interest, but also requires accurately localizing them. We believe our curated RD-Bench will serve as a helpful tool for both training and evaluation of models that aim at addressing this task.
>
> **Q3: "Comparison with ViperGPT"**
>
> Thanks for the suggestion. As suggested, we conducted additional experiments to compare our model with ViperGPT. It is worth noting that ViperGPT lacks support for the "localization" functionality and only generates textual answers. To address this limitation, we extracted the nouns from ViperGPT's answers and subsequently utilized GroundingDino for evaluation purposes. The resulting evaluation scores are presented in the table below (Due to the requirement of an API key from ChatGPT, we limited our evaluation to a subset of 100 samples to optimize cost efficiency. ):
>
>
> |             |  Task 1 | Task 2  | Task 3 | Task 4 |
> |----------|----------|----------|----------|----------|
> |             |   goal-orienated | all obejcts  | specific object|object w certain attribute |
> |      ViperGPT     |   10.37   |  13.43 | 23.64    |  16.21   |
> |      DetGPT      |   58.96   |   59.02  |   61.63  |  60.32   |
>
> We find that **ViperGPT does not follow the logic of “exhaustively detect all objects in an image, then leverage LLMs to determine the objects of interest”**; instead, it first searches for objects words (nouns) that exist in the instruction, then locate the object patches in the image.  We list a few examples of generated code and the final textual result, which shows the drawbacks of ViperGPT:
> - For some abstract instructions, ViperGPT cannot correctly understand and find the object of interest, instead, it uses the abstract word “danger” for detection. (example 1)
> - ViperGPT cannot combine the visual information for correct reasoning, it often returns the objects that are irrelevant to this image. (example 2)
>
> Example 1 from ViperGPT:
> - Input image: an image of a building with cracks on the wall;
> - Input text: From the image, find the potential danger to residents.
> - Generated code:
>
> ```
> def execute_command(image):
>     image_patch = ImagePatch(image)
>     danger_patches = image_patch.find("danger")
>     danger_patches.sort(key=lambda x: x.compute_depth())
>     return danger_patches[0]
> Final result: None
> ```
>
> Example 2 from ViperGPT:
>
> - Input image: an image of a kitchen with a refrigerator;
> - Input text:  Where can I find a cold beer from the image?
> - Generated code:
>
> ```
> def execute_command(image):
>     image_patch = ImagePatch(image)
>     beer_patches = image_patch.find("beer")
>     cold_beer_patches = [patch for patch in beer_patches if patch.verify_property("beer”, “cold”)]
>     if cold_beer_patches:
>         return cold_beer_patches[0].simple_query("Where can I find it?")
>     else:
>         return image_patch.llm_query("Where can I find a cold beer?")
> Final result: Most convenience stores, grocery stores, and liquor stores will sell cold beer. Additionally, many bars and restaurants will serve cold beer.
> ```
>
> We wish to clarify the following functions used in ViperGPT:
> - image_patch.find(object_name): detects the objects given their names;
> - image_patch.sort(key=lambda x: x.compute_depth()): sort the patches by their depths;
> - image_patch.verify_property(object_name, attribute) : determines whether the object satisfy the given property.
>
>
> We wish to note that **even if the logic follows “exhaustively detect all objects in an image, then leverage LLMs to determine the objects of interest”, it is challenging to accurately identify the target objects.** The inherent limitation with this paradigm is that the open-vocabulary object detectors require the object names as inputs for bounding box prediction.
> As an example, if the input image is an old building, the user instruction is "find the potential danger to residents", and the target object is “cracks on the wall”, it is questionable whether  “cracks on the wall” should be identified as a salient object in the first place. In order to truly extensively identify all objects, there must be a giant list containing all possible objects that may appear, and the detector needs to make predictions for each of them during inference, which is neither realistic nor practical. On the other hand, our DetGPT is able to flexibly detect the target objects via reasoning based on user instruction.
>
> **Q4: "Theoretical contribution compared with MiniGPT4"**
>
> Thanks for your feedback. We would like to first emphasize that our primary theoretical contribution is to verify that, through the 2nd-stage instruction tuning, VLMs can accurately identify objects based on high-level user instructions.  In contrast, MiniGPT4 struggles to find objects of interest and tends to output redundant objects (shown in Figure 4).  After being combined with an open-vocabulary detector, DetGPT is able to accurately localize objects of interest given high-level user instructions.
>
> Furthermore, the objective of our paper is to introduce the novel task of reasoning-based object detection. This task represents a significant departure from previous research, and we sincerely appreciate your recognition of its interesting nature and potential in application. To facilitate future research in this direction, we curate a benchmark to help both training and evaluation.
>
> **Q5: Reproducibility**
>
> Thanks for the feedback. We have included the prompts, and implementation details in our paper, and uploaded code as supplementary materials to this submission. We assure we will make the trained model, training/evaluation dataset, and codes publicly available upon acceptation.  As acknowledged by reviewer MiAP, our prompt details in the appendix help with reproducibility, and the RD-Bench dataset is a resource contribution to the community. With the released codes and datasets, it is easy to reproduce our results.
>
> Overall, many thanks for your insightful points and suggestions.  We have added the experiments accordingly. We hope our answers have addressed your concerns. If you have any further questions, we are happy to address them.

---

### Meta-Review · Area_Chair_YzV9 · 2023-09-16

**Recommendation:** 4

**Metareview:**

The new task of using reasoning for Indirect queries in the task of object detection is very interesting. The novelty of the task that the paper addresses is exciting. The authors have resolved many of the reviewers' questions with the rebuttal. I hope that these results will be properly reflected in the camera-ready version. Reviewer Hwbg makes a good point, but I don't think it is practical to cover the various data sets proposed by the reviewer. It is important that the paper is validated to be at least adequate to the claims made in the paper.

Pros:
The task is new and unique.
The paper is clearly written.
The results sound convincing.

Cons:
Many new results were presented during the rebuttal.

---

### Decision · Program_Chairs · 2023-10-07

**Decision:**

Accept-Main

**Comment:**

The new task of using reasoning for Indirect queries in the task of object detection is very interesting. The novelty of the task that the paper addresses is exciting. The authors have resolved many of the reviewers' questions with the rebuttal. I hope that these results will be properly reflected in the camera-ready version. Reviewer Hwbg makes a good point, but I don't think it is practical to cover the various data sets proposed by the reviewer. It is important that the paper is validated to be at least adequate to the claims made in the paper.

Pros:
The task is new and unique.
The paper is clearly written.
The results sound convincing.

Cons:
Many new results were presented during the rebuttal.